# Feature Map Similarity Reduction in Convolutional Neural Networks

**Zakariae Belmekki**                    *zakariae.belmekki@cranfield.ac.uk*
*Centre for Computational Engineering Sciences*
*Cranfield University*
*EstiaR*
*ESTIA-Institute of Technology*

**Jun Li**                    *jun.li@cranfield.ac.uk*
*Centre for Computational Engineering Sciences*
*Cranfield University*

**Patrick Reuter**                    *preuter@labri.fr*
*Inria*
*Univ. Bordeaux*
*EstiaR*
*ESTIA-Institute of Technology*

**David Antonio Gómez Jáuregui**                    *d.gomez@estia.fr*
*EstiaR*
*ESTIA-Institute of Technology*
*Univ. Bordeaux*

**Karl Jenkins**                    *k.w.jenkins@cranfield.ac.uk*
*Centre for Computational Engineering Sciences*
*Cranfield University*

## Abstract

It has been observed that Convolutional Neural Networks (CNNs) suffer from redundancy in feature maps, leading to inefficient capacity utilization. Efforts to address this issue have largely focused on kernel orthogonality method. In this work, we theoretically and empirically demonstrate that kernel orthogonality does not necessarily lead to a reduction in feature map redundancy. Based on this analysis, we propose the Convolutional Similarity method to reduce feature map similarity, independently of the CNN's input. The Convolutional Similarity can be minimized as either a regularization term or an iterative initialization method. Experimental results show that minimizing Convolutional Similarity not only improves classification accuracy but also accelerates convergence. Furthermore, our method enables the use of significantly smaller models to achieve the same level of performance, promoting a more efficient use of model capacity. Future work will focus on coupling the iterative initialization method with the optimization momentum term and examining the method's impact on generative frameworks.

## 1 Introduction

Convolutional Neural Networks (CNNs) are widely regarded as a powerful class of models in Deep Learning. Over the years, numerous advanced convolutional architectures have been developed, achieving remarkable performance in tasks such as classification (He et al., 2015; Simonyan and Zisserman, 2015; Krizhevsky

et al., 2012), segmentation (Ronneberger et al., 2015), and data generation (Sauer et al., 2023; Karras et al., 2021). Despite their success, the capacity and performance of CNNs are often hindered by the presence of information redundancy in trained models (Wang et al., 2020; Rodríguez et al., 2017; Cogswell et al., 2016; Jaderberg et al., 2014), evidenced in both feature maps (Cogswell et al., 2016) and kernels (Denil et al., 2013; Wang et al., 2020).

Redundancy in CNNs has been explored in various studies, primarily through kernel orthogonality (Kahatapitiya and Rodrigo, 2021; Liang et al., 2020; Cogswell et al., 2016; Denil et al., 2013). However, no existing literature focuses on the impact of feature kernel, feature map and their relationship on the redundancy, including how kernel orthogonality affects feature map orthogonality. Furthermore, minimizing redundancy to enhance the efficient utilization of CNN capacity remains an underexplored area (Rodríguez et al., 2017; Cogswell et al., 2016). Without a redundancy-minimizing mechanism, CNNs are unable to intentionally learn non-redundant features.

A common approach to reducing similarity between two feature maps, $F_1$ and $F_2$, is to orthogonalize them by defining an objective function based on vector inner product. However, this method may introduce significant computational overhead, particularly when dealing with large datasets. A more efficient solution is to reduce the inner product of $F_1$ and $F_2$ by leveraging the characteristics of feature kernels, which requires a theoretical analysis of the relationship between feature map similarity and kernel properties. This study aims to theoretically and experimentally analyze the relationship between feature map orthogonality and feature kernels in CNNs. Based on the derived theory, we propose a novel Convolutional Similarity loss function that minimizes feature map similarity either prior to or during model training. This minimization relies solely on feature kernels, making its computational complexity independent of input size, unlike traditional feature map decorrelation methods (Cogswell et al., 2016). Subsequent experiments demonstrate that minimizing Convolutional Similarity improves the prediction accuracy of CNNs, accelerates convergence, and reduces computational complexity.

The paper is organized as follows: Section 2 provides an overview of existing work in this area; Section 3 develops the theory for reducing feature map similarity in CNNs, beginning with an empirical identification of the limitations of kernel orthogonality in minimizing feature map similarity in Section 3.1, followed by the derivation of the Convolutional Similarity loss in Section 3.2. The theory is numerically validated, and then evaluated in Section 4; Finally, Section 5 concludes the work.

## 2  Related Work

Convolutional layers in CNNs often learn redundant features with correlated weights, leading to model inefficiency (Hssayni et al., 2023; Ding et al., 2021; Kahatapitiya and Rodrigo, 2021; Wang et al., 2021; Liang et al., 2020). Various methods are available to reduce feature redundancy, including efficient architecture design, redundant weight pruning, and regularization to minimize redundancy in weights and activations. Wang et al. (2021) proposed a structural redundancy approach that prunes filters in selected layers with the most redundancy, resulting in more compact and efficient network architectures. Ding et al. (2021) introduced a novel Centripetal SGD (C-SGD) method that forces certain filters to become identical, enabling the pruning of these filters with minimal performance loss. Alternatively, Hssayni et al. (2023) developed an L1-sparsity optimization model for detecting and reducing unnecessary kernels. Cogswell et al. (2016) introduced a regularization method called DeCov to minimize activation covariance. While DeCov reduces redundancy and improves test accuracy, it operates solely on the outputs of the last fully connected layer rather than directly on convolutional feature maps. Its effectiveness is therefore dataset-dependent and does not guarantee the reduction of similarity in convolutional feature maps.

Several methods have been proposed to reduce redundant features through kernel orthogonality. Kahatapitiya and Rodrigo (2021) observed that convolutional layers often learn redundant features due to correlated filters and introduced the LinearConv layer. This approach learns a set of orthogonal filters and coefficients to linearly combine them, reducing redundancy and the number of parameters without compromising performance. Rodríguez et al. (2017) proposed a regularization method called OrthoReg to reduce positive correlation between convolutional kernels based on cosine similarity. However, this method primarily targets model parameter redundancy and overfitting rather than directly addressing feature redundancy. Wang et al.

(2020) proposed Orthogonal CNNs (OCNNs), further explored by Achour et al. (2022), to impose kernel orthogonality through linear transformation and regularization. This approach shows consistent performance improvement with various network architectures on various tasks. These approaches operate on kernels for redundancy reduction under the assumption that kernel orthogonality leads to feature map redundancy reduction, which we will show to be false. Subsequently, we propose a novel Convolutional Similarity loss function that minimizes feature map similarity based on the orthogonality operation on convolutional feature kernels.

## 3 Convolutional Similarity and Feature Map Orthogonality

A Convolutional Neural Network (CNN) often performs cross-correlation than convolution in its standard implementation such as PyTorch and Tensorflow for forward propagation, with three typical variants of padding:

1. **Full convolution/cross-correlation**: An $N-1$ padding is added to both sides of the input vector. For an input vector $X \in \mathbb{R}^M$ and a kernel $K \in \mathbb{R}^N$, the full convolution and cross-correlation are calculated as follows:

$$(X * K)[i] = \sum_{n=0}^{N-1} K[n] \cdot X[i-n] \tag{1a}$$

$$(X \circledast K)[i] = \sum_{n=0}^{N-1} K[n] \cdot X[i+n-N+1], i \in [0, M+N-2]. \tag{1b}$$

where $*$ and $\circledast$ denote the convolution and the cross-correlation operators respectively.

2. **Valid convolution/cross-correlation**: No padding is added to the input. The size of the output $F$, given an input vector $X \in \mathbb{R}^M$ and a kernel $K \in \mathbb{R}^N$, is $F \in \mathbb{R}^{M-N+1}$. The calculations are same, with $i \in [N-1, M-1]$.

3. **Same convolution/cross-correlation**: A padding of size $\frac{N-1}{2}$ is added to both sides of the input, resulting in feature maps with the same size as the input. The calculations are same, with $i \in [\frac{N-1}{2}, M-1+\frac{N-1}{2}]$.

This work assumes that the convolutional layers perform cross-correlation. The results are equally valid for convolution operations.

### 3.1 Problem Identification

Let us assess the impact of kernel similarity minimization on feature map similarity. Given the feature maps $F_1, F_2 \in \mathbb{R}^{M+N-1}$ resulting from full cross-correlation, with input $X \in \mathbb{R}^M$, kernels $K_1, K_2 \in \mathbb{R}^N$, and $i \in [0, M+N-2]$, one has

$$F_1[i] = (K_1 \circledast X)[i] = \sum_{n_1=0}^{N-1} K_1[n_1] \cdot X[n_1+i-N+1] \tag{2a}$$

$$F_2[i] = (K_2 \circledast X)[i] = \sum_{n_2=0}^{N-1} K_2[n_2] \cdot X[n_2+i-N+1]. \tag{2b}$$

The problem is to study the effect of minimizing kernel similarity, i.e., $\min_{K_1,K_2}\langle K_1, K_2 \rangle^2 \to 0$, on feature map similarity, i.e., $\langle F_1, F_2 \rangle$.

Each optimization experiment is run over a number of iterations for each input and kernel pair, with Adam and Stochastic Gradient Descent (SGD) optimizers (see Table 7 for the full configuration). Due to its

stochastic nature, each optimization is repeated 1000 episodes and the mean value is computed. The the learning rates and the number of iterations vary depending on the problem parameters, e.g., the kernel size. The kernel vectors $K_1$, $K_2$ and input vectors $X$ are sampled from uniform distributions, where $K_1, K_2 \sim \mathcal{U}(-1, 1)$ and $X \sim \mathcal{U}(0, 1)$, with a varied he kernel size $N$ and a fixed input size $M = 64$. The correlation between kernel similarity and feature map similarity, the frequency of feature map similarity reduction (i.e., the ratio of number of decreases to total episodes), and the percentage change in feature map similarity when it increases or decreases are measured. The results are presented in Table 1

| N | Optimiser | Correlation | | Reduction Frequency (%) | Decrease (%) | | Increase (%) | |
|---|---|---|---|---|---|---|---|---|
| | | mean | std | | mean | std | mean | std |
| 3 | Adam | 0.67 | 0.28 | 91.8 | 92.95 | 15.24 | $14.12 \times 10^3$ | $71.12 \times 10^3$ |
| | SGD | 0.35 | 0.85 | 67.5 | 78.75 | 24.71 | $13.82 \times 10^4$ | $10.60 \times 10^5$ |
| 9 | Adam | 0.54 | 0.33 | 82 | 86.12 | 21.14 | $17.52 \times 10^4$ | $20.85 \times 10^5$ |
| | SGD | 0.25 | 0.87 | 60.7 | 67.82 | 29.42 | $55.76 \times 10^4$ | $87.15 \times 10^5$ |
| 16 | Adam | 0.47 | 0.36 | 77.6 | 81.88 | 23.48 | $17.10 \times 10^4$ | $20.50 \times 10^5$ |
| | SGD | 0.54 | 0.33 | 71.6 | 69.84 | 28.19 | $97.62 \times 10^2$ | $74.75 \times 10^3$ |

Table 1: The effect of minimizing kernel similarity on feature map similarity.

The table shows no significant linear dependency between kernel similarity and feature map similarity, as indicated by the low mean and high standard deviation of correlations. The results reveal that kernel orthogonality can lead to either a significant decrease or increase in feature map similarity, depending on the setting. For example, while the reduction frequency and the decrease percentage are high with $N = 3$ and Adam optimizer, the order of magnitude of the increase percentage is large. In the next section, we theoretically validate this observation and propose a method for reducing feature map similarity.

### 3.2 Derivation of the Convolutional Similarity

Given $F_1$ and $F_2$ in Equations 2, the problem is formulated as finding a solution to achieve the orthogonality, $\langle F_1, F_2 \rangle = 0$, with regard to the kernels $K_1$ and $K_2$. Here we will prove that the inner product of feature maps can be expressed as the inner product of the auto-correlation of the input $X$, clipped to the range $[\mathbf{1 - N, N - 1}]$, and the full cross-correlation of $K_1$ and $K_2$, as shown in Equation 3, deriving a sufficient condition on kernels for for achieving feature map orthogonality.

$$\langle F_1, F_2 \rangle = \langle (K_1 \circledast K_2), (X \circledast X)_{[1-N,N-1]} \rangle. \tag{3}$$

From Equations 2, it follows that:

$$\langle F_1, F_2 \rangle = \sum_{i=0}^{M+N-2} F_1[i] \cdot F_2[i] = \sum_{n_1=0}^{N-1} \sum_{n_2=0}^{N-1} K_1[n_1] \cdot K_2[n_2] \sum_{i=0}^{M+N-2} X[i + n_2 - N + 1] \cdot X[i + n_1 - N + 1]. \tag{4}$$

For simplicity, let $i = i + n_1 - N + 1$. Then:

$$\langle F_1, F_2 \rangle = \sum_{n_1=0}^{N-1} \sum_{n_2=0}^{N-1} K_1[n_1] \cdot K_2[n_2] \sum_{i=1-N+n_1}^{M-1+n_1} X[i + n_2 - n_1] \cdot X[i]. \tag{5}$$

Given the zero-padding, the equation can be simplified to:

$$\langle F_1, F_2 \rangle = \sum_{n_1=0}^{N-1} \sum_{n_2=0}^{N-1} K_1[n_1] \cdot K_2[n_2] \sum_{i=0}^{M-1} X[i + n_2 - n_1] \cdot X[i].$$

Let $n_2 = n_2 - n_1$, then:

$$\langle F_1, F_2 \rangle = \sum_{n_1=0}^{N-1} \sum_{n_2=-n_1}^{N-1-n_1} K_1[n_1] \cdot K_2[n_1 + n_2] \sum_{i=0}^{M-1} X[i + n_2] \cdot X[i]$$

$$= \sum_{n_1=0}^{N-1} \sum_{n_2=-n_1}^{N-1-n_1} K_1[n_1] \cdot K_2[n_1 + n_2] \cdot (X \circledast X)[n_2]. \tag{6}$$

With

$$A = \sum_{n_1=0}^{N-1} \sum_{n_2=N-n_1}^{N-1} K_1[n_1] \cdot K_2[n_1 + n_2] \cdot (X \circledast X)[n_2] = 0,$$

$$B = \sum_{n_1=0}^{N-2} \sum_{n_2=1-N}^{-n_1-1} K_1[n_1] \cdot K_2[n_1 + n_2] \cdot (X \circledast X)[n_2] = 0,$$

where $K_2$ values are out of range and set to zeros, we obtain Equation 3:

$$\langle F_1, F_2 \rangle = \langle F_1, F_2 \rangle + A + B = \sum_{n_2=1-N}^{N-1} \sum_{n_1=0}^{N-1} K_1[n_1] \cdot K_2[n_1 + n_2] \cdot (X \circledast X)[n_2]$$

$$= \sum_{n_2=1-N}^{N-1} (K_1 \circledast K_2)[n_2] \cdot (X \circledast X)[n_2] = \langle (K_1 \circledast K_2), (X \circledast X)_{[1-N, N-1]} \rangle.$$

Then, a trivial sufficient condition on the kernels for the orthogonality of $F_1$ and $F_2$ can be derived in the case of the full cross-correlation:

$$S = \{n \in [1-N, N-1] | (K_1 \circledast K_2)[n] = 0\}. \tag{7}$$

We note that the same result can be derived using Parseval's and the Convolution theorems.

In an optimization scenario, the condition can be achieved by minimizing the Convolutional Similarity:

$$\sum_{n=1-N}^{N-1} (K_1 \circledast K_2)^2[n]. \tag{8}$$

In its general form, for feature maps $S > 2$ and input channels $C > 1$, the Convolutional Similarity $L_{CS}$ is written as:

$$L_{CS} = \sum_{i=0}^{S-1} \sum_{\substack{j=0 \\ j \neq i}}^{S-1} \sum_{c_1=0}^{C-1} \sum_{c_2=0}^{C-1} \sum_{n=1-N}^{N-1} (K_i[c_1] \circledast K_j[c_2])^2[n], \tag{9}$$

where $K_i[c_1]$ denotes the $c_1$-th channel of the $i$-th kernel.

One may see that the components of Equation 8 are symmetric with recurring terms. Therefore, $L_{CS}$ can be simplified to:

$$L_{CS} = \sum_{i=0}^{S-2} \sum_{j>i}^{S-1} \sum_{c_1=0}^{C-1} \sum_{c_2=0}^{C-1} \sum_{n=1-N}^{N-1} (K_i[c_1] \circledast K_j[c_2])^2[n]. \tag{10}$$

Similarly, feature map orthogonality in the case of the full convolution operation can also be achieved.

Given Equation 3, we have:

$$\langle F_1, F_2 \rangle = \sum_{n=1-N}^{N-1} (X \circledast X)[n] \cdot (K_1 \circledast K_2)[n]$$

$$= (X \circledast X)[0] \cdot (K_1 \circledast K_2)[0] + \sum_{\substack{n=1-N \\ n \neq 0}}^{N-1} (X \circledast X)[n] \cdot (K_1 \circledast K_2)[n]$$

$$= \|X\|_2^2 \cdot \langle K_1, K_2 \rangle + \sum_{\substack{n=1-N \\ n \neq 0}}^{N-1} (X \circledast X)[n] \cdot (K_1 \circledast K_2)[n].$$

Allowing $\langle K_1, K_2 \rangle = 0$, we have:

$$\langle F_1, F_2 \rangle = \sum_{\substack{n=1-N \\ n \neq 0}}^{N-1} (X \circledast X)[n] \cdot (K_1 \circledast K_2)[n]. \tag{11}$$

This shows that kernel orthogonality does not lead to feature map orthogonality, consistent with the results from Section 3.1. Moreover, the Convolutional Similarity approach differs from Orthogonal Convolutional Neural Networks (OCNN) (Wang et al., 2020) in two ways. First, OCNN applies kernel orthogonality to reduce feature map redundancy, whereas we prove that no correlation exists between the two. Second, Convolutional Similarity performs the full cross-correlation across kernel channels, while OCNN performs cross-correlation within channels. The mathematical deduction establishes a sufficient condition on kernels for feature map orthogonality with full cross-correlation. The extension to valid and same convolution/cross-correlation based an approximate approach is given in Appendix 6.1.

## 4 Experiments and Results

### 4.1 Numerical Validation

Here we minimize the Convolutional Similarity, i.e., $\min_{K_1, K_2} \|(K_1 \circledast K_2)\|_2^2$, to validate its effect on feature map orthogonality by performing the same experiments conducted in Section 3.1. The results are presented in Table 2 (See the optimization hyper-parameters for each $N$ in Table 8)

| N | Optimiser | Correlation | |
|---|---|---|---|
| | | mean | std |
| 3 | Adam | 0.80 | 0.27 |
| | SGD | 0.90 | 0.21 |
| 9 | Adam | 0.78 | 0.27 |
| | SGD | 0.85 | 0.25 |
| 16 | Adam | 0.71 | 0.30 |
| | SGD | 0.86 | 0.25 |

Table 2: The effect of minimizing Convolutional Similarity on feature map similarity.

The table shows that Convolutional Similarity has a high and stable correlation with feature map similarity, in contrast to the results in Table 1. Table 3 further shows that, in all cases, minimizing Convolutional Similarity consistently decreases feature map similarity, achieving near 100% orthogonality.

| N | Optimiser | Reduction Frequency (%) | Decrease (%) | | Increase (%) | |
|---|---|---|---|---|---|---|
| | | | mean | std | mean | std |
| 3 | Adam | 100 | 100 | 0.0 | 0 | 0 |
| | SGD | 100 | 99.91 | 1.28 | 0 | 0 |
| 9 | Adam | 100 | 99.98 | 0.23 | 0 | 0 |
| | SGD | 100 | 99.68 | 3.21 | 0 | 0 |
| 16 | Adam | 100 | 99.95 | 1.26 | 0 | 0 |
| | SGD | 100 | 99.78 | 3.04 | 0 | 0 |

Table 3: The effect of totally minimising Convolutional Similarity, on feature map similarity.

## 4.2 Convolutional Similarity Minimization Algorithms and Experiments on Shallow CNNs

Convolutional Similarity Minimization can be performed either before training as an iterative initialization scheme as shown in Algorithm 1, or during training as a regularisation method as shown in Algorithm 2.

---

**Algorithm 1** Convolutional Similarity minimization as an iterative initialization scheme. $\theta$ are the parameters of the model and $K^{(j)}$ are the kernels of the $j$-th convolutional layer. $L_{task}$ and $L_{CS}$ are the training task loss function and the Convolutional Similarity loss function, respectively. Before the model is trained, $L_{CS}$ is minimized for a number of iterations $I$.

**for** $I$ iterations **do**
    One optimization step of the Convolutional Similarity minimization:

$$\nabla_\theta \sum_{j=0}^{M-1} L_{CS}(K^{(j)})$$

**end for**
**for** training epochs **do**
    One training epoch of the model:

$$\nabla_\theta L_{task}$$

**end for**

---

**Algorithm 2** Convolutional Similarity minimization as a regularization term. $\theta$ are the parameters of the model and $K^{(j)}$ are the kernels of the $j$-th convolutional layer. $L_{task}$ and $L_{CS}$ are the training task loss function and the Convolutional Similarity loss function, respectively. $L_{CS}$ is minimized during the training of the model and is used as a regularisation term with a weighting factor $\beta$.

**for** training epochs **do**
    Convolutional Similarity is minimized with every epoch

$$\nabla_\theta \left( L_{task} + \beta \cdot \sum_{j=0}^{M-1} L_{CS}(K^{(j)}) \right)$$

**end for**

---

Experiments are conducted with two shallow CNNs on CIFAR-10 (Krizhevsky and Hinton, 2009), a dataset containing 50,000 training images and 10,000 test images for classification. Both of the aforementioned methods for Convolutional Similarity minimization are applied. Further experiments using the deep ResNet18 model He et al. (2015) on the same dataset with an arbitrary number of padding are presented in the following section.

| CNN1 |
|---|
| Conv2d(3, 64, 3, 2) |
| BatchNorm2d(64) |
| LeakyReLU(0.2) |
| MaxPool2d(2, 2) |
| Conv2d(64, 64, 3, 2) |
| BatchNorm2d(64) |
| LeakyReLU(0.2) |
| MaxPool2d(2, 2) |
| Conv2d(64, 64, 3, 2) |
| BatchNorm2d(64) |
| LeakyReLU(0.2) |
| MaxPool2d(2, 2) |
| Conv2d(64, 64, 3, 2) |
| BatchNorm2d(64) |
| LeakyReLU(0.2) |
| MaxPool2d(2, 2) |
| Linear(576, 10) |

| CNN2 |
|---|
| Conv2d(3, 128, 3, 2) |
| BatchNorm2d(128) |
| LeakyReLU(0.2) |
| MaxPool2d(2, 2) |
| Conv2d(128, 128, 3, 2) |
| BatchNorm2d(128) |
| LeakyReLU(0.2) |
| MaxPool2d(2, 2) |
| Conv2d(128, 128, 3, 2) |
| BatchNorm2d(128) |
| LeakyReLU(0.2) |
| MaxPool2d(2, 2) |
| Conv2d(128, 128, 3, 2) |
| BatchNorm2d(128) |
| LeakyReLU(0.2) |
| MaxPool2d(2, 2) |
| Linear(1152, 10) |

Table 4: The configurations of CNN1 and CNN2.

The two shallow CNNs share the same architecture but CNN2 has twice as many feature maps per layer and 3.86 times more trainable parameters overall (118,858 vs 458,890), as detailed in Table 4. In regularization method, the SGD optimiser with a learning rate of 0.01 is used for Convolutional Similarity minimization, while in iterative initialization method, the Adam optimiser with a learning rate of 0.001 is used to minimize the Convolutional Similarity loss prior to training and the SGD optimiser with a learning rate of 0.01 is used to minimize the Cross Entropy loss during training. The models are trained with a batch size of 512 for 100 epochs. Table 5 presents the top test accuracies for the baseline models, the models trained with $I = 500$ iterations prior to training, and the models trained with the regularization term and weight $\beta = 0.001$. Both methods based on Convolutional Similarity improve prediction accuracy compared to the baseline models, with the pre-training approach achieving the best performance.

| Model | Test Accuracy (%) |
|---|---|
| CNN1 (baseline) | 74.66 |
| CNN1 ($I = 500$) | **79.16** |
| CNN1 ($\beta = 0.001$) | 77.16 |
| CNN2 (baseline) | 76.87 |
| CNN2 ($I = 500$) | **82.33** |
| CNN2 ($\beta = 0.001$) | 80.00 |

Table 5: Top test accuracies for CNN1 and CNN2.

In addition, despite having 3.86 times fewer trainable parameters, CNN1 trained with Convolutional Similarity outperforms the CNN2 baseline. Further testing reveals that the CNN2 baseline requires 512 filters per layer (seven times more) with 7,143,946 trainable parameters (60.10 times more) to achieve an accuracy of 79.78%, matching the performance of CNN1 with $I = 500$.

Experiments also show that the iterative initialization approach introduces less computational overhead compared to the regularization method. While the latter incurs 9,800 additional optimization iterations (98 per epoch), the iterative initialization method only adds $I = 500$ iterations more. Furthermore, the model can be reused for different tasks after a one-off pre-training. However, in iterative initialization a large learning rate can lead to catastrophic forgetting Kirkpatrick et al. (2017), where the model loses the weights

learned. Given the Convolutional Similarity function being convex, a properly tuned learning rate ensures that the Convolutional Similarity loss remains low at training phase, as shown in Figure 1.

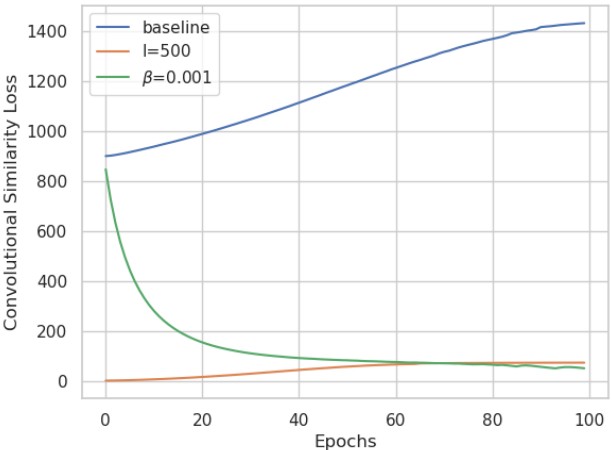

Figure 1: Convolutional Similarity loss curves for (a) the baseline, (b) $I = 500$, and (c) $\beta = 0.001$.

Figures 2 and 3 show the evolution of the loss and the accuracy during training for both models, indicating that the iterative initialization method accelerates convergence. In Figure 2, the classification loss decreases fastest for the models with $I = 500$, while more slowly for the models trained with $\beta = 0.001$, potentially due to the Convolutional Similarity loss as a regularisation term distorting the classification loss. This becomes more apparent when the classification loss is low, therefore, the gradients of the Convolutional Similarity are more likely to cause oscillations. Models trained with either $I = 500$ or $\beta = 0.001$ achieve higher train accuracy and reach near 100% earlier than the baseline models, as shown in Figure 3. The oscillations in Figure 2 are similarly reflected in the train accuracy in Figure 3.

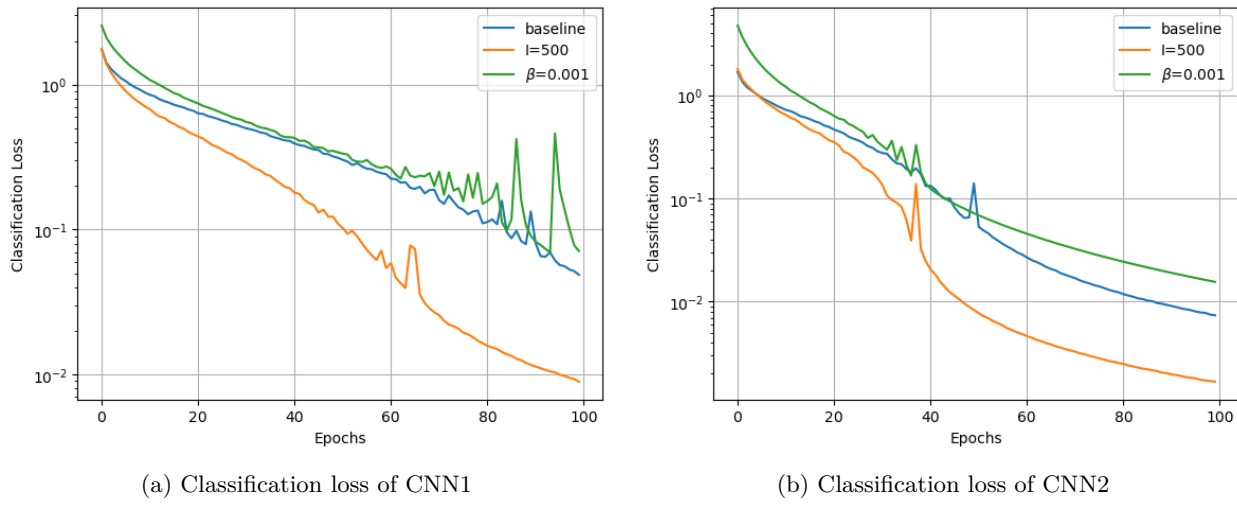

(a) Classification loss of CNN1         (b) Classification loss of CNN2

Figure 2: The classification loss evolution for CNN1 and CNN2 on a logarithmic scale.

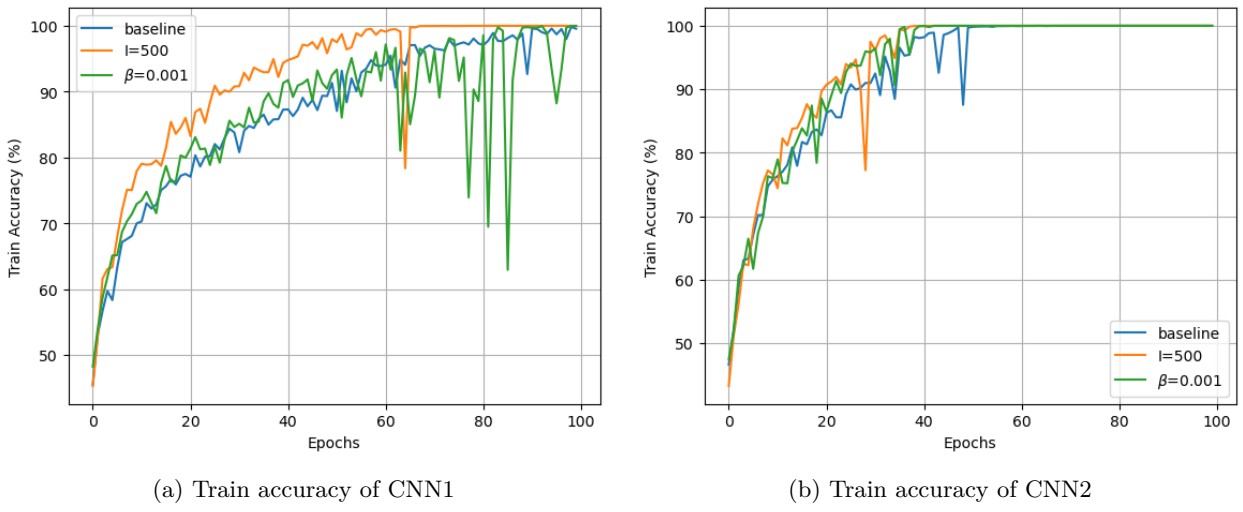

(a) Train accuracy of CNN1          (b) Train accuracy of CNN2

Figure 3: The train accuracies of CNN1 and CNN2.

### 4.3 Convolutional Similarity Minimization Experiments on a Deep CNN

Convolutional Similarity Minimization is further evaluated on a deep model, ResNet18, using the same dataset. The model is pre-trained using the iterative initialization method with $I = 200$ and then trained for 50 epochs using SGD with various learning rates and batch sizes. The testing results, presented in Figure 4 show that the models employing Convolutional Similarity Minimization outperform the baseline models, with significant accuracy improvements in certain cases (e.g., $lr = 0.001, batchsize = 128 \rightarrow \Delta Accuracy = 20\%$). These findings confirm the robustness of the Convolutional Similarity method.

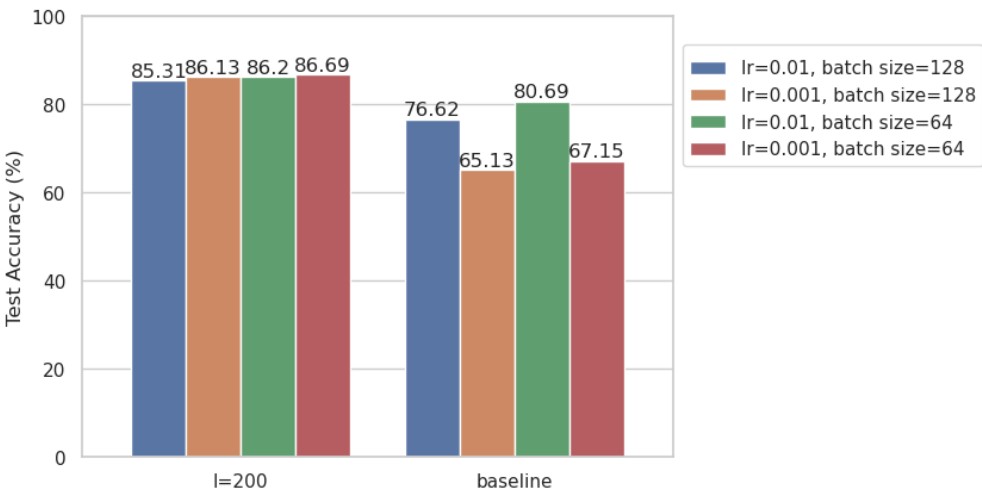

Figure 4: Accuracies of the ResNet18 model with different configurations, trained with and without Convolutional Similarity Minimization.

To determine the number of iterations $I$ for best model performance, the ResNet18 model is subsequently trained for 50 epochs using a step-wise search with $I \in \{0, 10, 20, 40, 60, 80, 100, 120, 140, 160, 180, 200\}$, a learning rate of 0.01 and a batch size of 64. The results, shown in Figure 5, indicate that most accuracy improvements occur within $I < 60$, and a model with $I$ as low as 10 yet produces a good performance. This finding is valuable for deeper model with costly Convolutional Similarity gradient computation.

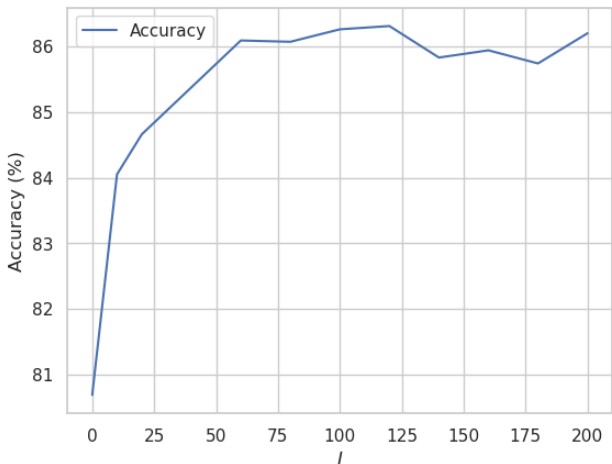

Figure 5: Test accuracy variation of ResNet18 model according to iterations $I$.

A limitation of the iterative initialization method is its reduced effectiveness when the task loss is minimized using optimizers with momentum or a large learning rate. It is observed that optimizers with momentum cause an increase of Convolutional Similarity loss previously minimized, a phenomenon similar to that reported in (Heo et al., 2021). Moreover, the large learning rates push the weights outside of the region where Convolutional Similarity is minimal. While optimizers with momentum work better with Convolutional Similarity minimization as a regularisation method, this approach introduces additional computational overhead and causes oscillations near the end of training due to small gradient values. A potential improvement is to reduce $\beta$ in proportion to the magnitude of the task loss.

## 5  Conclusion and Future Work

This study introduces Convolutional Similarity, a novel loss term for Convolutional Neural Networks (CNNs), derived from a theoretical analysis of feature map orthogonality to reduce information redundancy retained in CNNs. The Convolutional Similarity can be minimized as either a regularization term or an iterative initialization scheme. Base on the experiments, the minimization of Convolutional Similarity not only reduces the feature map similarity independent of the CNN input, but also leads to an improved prediction accuracy and accelerate convergence of CNN models. Additionally, the method enables significantly smaller models to achieve performance comparable to larger models that do not use it. This research also demonstrates, both theoretically and empirically, that kernel orthogonality does not necessarily reduce feature map similarity, contrary to claims in some literature. A limitation of the iterative initialization method is its reduced effectiveness due to optimization momentum and large learning rates. A potential solution is to either search for an optimal learning rate or use Convolutional Similarity as a regularization term, and then determine the optimal weight. Future work will focus on coupling the iterative initialization method with the optimization momentum term and examining the method's impact on generative frameworks such as Generative Adversarial Networks.

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

## 6 Appendix

### 6.1 Extension to Valid and Same Convolution/Cross-correlation

We have established a sufficient condition on kernels for feature map orthogonality, with full cross-correlations. In this section, an approximative approach is adopted to extend the use of Convolutional Similarity minimization to same and valid cross-correlation/convolution scenarios (e.g., used in Residual Neural Networks He et al. (2015). Given the general definition of the cross-correlation with the padding $P \in [0, N-1]$ and $i \in [N-1-P, M-1+P]$, one has:

$$F_1[i] = (X \circledast K_1)[i] = \sum_{n_1=0}^{N-1} K_1[n_1] \cdot X[i + n_1 - N + 1], \tag{12}$$

$$F_2[i] = (X \circledast K_2)[i] = \sum_{n_2=0}^{N-1} K_2[n_2] \cdot X[i + n_2 - N + 1], \tag{13}$$

$$\langle F_1, F_2 \rangle = \sum_{i=N-1-P}^{M-1+P} F_1[i] \cdot F_2[i]. \tag{14}$$

where the valid, full and same cross-correlation correspond to $P = 0$, $P = N - 1$ and $P = \frac{N-1}{2}$ respectively. With the substitution of $i = n_1 + i - N + 1$, one has:

$$\langle F_1, F_2 \rangle = \sum_{n_1=0}^{N-1} \sum_{n_2=0}^{N-1} K_1[n_1] \cdot K_2[n_2] \sum_{i=n_1-P}^{M-N+P+n_1} X[i + n_2 - n_1] \cdot X[i] \tag{15}$$

$$= -A - B + \sum_{n_1=0}^{N-1} \sum_{n_2=0}^{N-1} K_1[n_1] \cdot K_2[n_2] \sum_{i=1-N+n_1}^{M-1+n_1} X[i + n_2 - n_1] \cdot X[i], \tag{16}$$

with

$$A = \sum_{n_1=0}^{N-1} \sum_{n_2=0}^{N-1} K_1[n_1] \cdot K_2[n_2] \sum_{i=n_1+1-N}^{n_1-P-1} X[i + n_2 - n_1] \cdot X[i], \tag{17}$$

$$B = \sum_{n_1=0}^{N-1} \sum_{n_2=0}^{N-1} K_1[n_1] \cdot K_2[n_2] \sum_{i=M-N+P+n_1+1}^{M-1+n_1} X[i + n_2 - n_1] \cdot X[i]. \tag{18}$$

Using the same procedure as for Equation 3, one has:

$$\langle F_1, F_2 \rangle = \langle (K_1 \circledast K_2), (X \circledast X)_{[1-N, N-1]} \rangle - A - B. \tag{19}$$

Where $A$ and $B$ have $N \cdot N \cdot (N - P - 1)$ terms while $\langle (K_1 \circledast K_2), (X \circledast X)_{[1-N, N-1]} \rangle$ $N \cdot N \cdot (M + N - 1)$ terms. Given $N << M$ (The kernel dimension size is often significantly smaller than the input dimension) and $P \leq N - 1$, there is a high probability that

$$\langle F_1, F_2 \rangle \approx \langle (K_1 \circledast K_2), (X \circledast X)_{[1-N, N-1]} \rangle. \tag{20}$$

As $P$ approaches to $N - 1$, the two sides of Equation 20 become increasingly close. When $P = N - 1$, one has

$$\langle F_1, F_2 \rangle = \langle (K_1 \circledast K_2), (X \circledast X)_{[1-N, N-1]} \rangle. \tag{21}$$

The approximation is verified numerically by performing the same numerical tests for the valid cross-correlation as in Section 4.1, as shown in Table 6. Similar to Table 1, the results show that minimising

Convolutional Similarity is effective for arbitrary $P$ values. The correlations between feature map similarity and Convolutional Similarity are significantly higher than those between feature map similarity and kernel similarity as presented in Table 1. In all cases, feature map similarity is guaranteed to decrease and approach orthogonality. This result is further confirmed in Section 4.3, where Convolutional Similarity is minimized for a deep CNN that does not use the full cross-correlation. However, in some rare cases, minimizing Convolutional Similarity can have an unpredictable effect on feature map similarity, as shown in the Increase column, similar to the effect of minimizing kernel similarity in Table 1. Nonetheless, this phenomenon is significantly less pronounced, with a probability of 0.014 in the worst case, as observed in the table.

| N | Optimiser | Correlation | | Reduction Frequency (%) | Decrease (%) | | Increase (%) | |
|---|---|---|---|---|---|---|---|---|
| | | mean | std | | mean | std | mean | std |
| 3 | Adam | 0.798 | 0.272 | 100 | 99.99 | 0.02 | 0 | 0 |
| | SGD | 0.90 | 0.19 | 99.6 | 99.86 | 1.34 | 216.04 | 154.33 |
| 9 | Adam | 0.751 | 0.296 | 100 | 99.96 | 0.385 | 0 | 0 |
| | SGD | 0.86 | 0.23 | 99.5 | 99.57 | 3.77 | 3336.06 | 5322.40 |
| 16 | Adam | 0.77 | 0.28 | 99.9 | 99.80 | 2.49 | 688.54 | 0 |
| | SGD | 0.851 | 0.23 | 98.6 | 98.68 | 8.20 | $10.90 \times 10^4$ | $40.35 \times 10^4$ |

Table 6: The effect of minimising Convolutional Similarity on feature map similarity. See Table 9 for optimization hyper-parameters for each $N$.

## 6.2 Numerical Validation Experiments Configurations

| N | Optimizer | Learning Rate | Number of Iterations |
|---|---|---|---|
| 3 | Adam | 0.1 | 250 |
| | SGD | 0.1 | 250 |
| 9 | Adam | 0.1 | 250 |
| | SGD | 0.1 | 250 |
| 16 | Adam | 0.1 | 250 |
| | SGD | 0.1 | 300 |

Table 7: Optimization hyperparameters for kernel cosine similarity minimization experiments.

| N | Optimizer | Learning Rate | Number of Iterations |
|---|---|---|---|
| 3 | Adam | 0.1 | 300 |
| | SGD | 0.2 | 350 |
| 9 | Adam | 0.1 | 400 |
| | SGD | 0.07 | 550 |
| 16 | Adam | 0.2 | 450 |
| | SGD | 0.035 | 1500 |

Table 8: Optimization hyperparameters for Convolutional Similarity optimization experiments.

| N | Optimizer | Learning Rate | Number of Iterations |
|---|---|---|---|
| 3 | Adam | 0.1 | 300 |
| | SGD | 0.1 | 300 |
| 9 | Adam | 0.1 | 300 |
| | SGD | 0.05 | 400 |
| 16 | Adam | 0.05 | 500 |
| | SGD | 0.01 | 700 |

Table 9: Optimization hyperparameters for kernel cosine similarity minimization experiments.

