# OpenReview forum: "Feature Map Similarity Reduction in Convolutional Neural Networks"
_TMLR — Rejected by TMLR_

### Review · Reviewer_VZZT · 2024-12-05

**Summary Of Contributions:**

The paper can easily fail an undergraduate signal processing exam. There are so many fundamental issues in the paper.

**Audience:**

No

**Claims And Evidence:**

No

**Requested Changes:**

Please reject this paper without further consideration.

**Strengths And Weaknesses:**

No strength, only weaknesses.

Let's talk about some high-level issues before technical problems.
1. First of all, kernel orthogonality was NOT supposed to reduce feature map redundancy (and I am not aware of such claim before). The purposes of kernel orthogonality are for robustness (due to its Lipschitz property), invertibility, and regularization. There is another line of works to reduce feature map redundancy, such as network deconvolution (ICLR 2020) or simply batch/layer normalization. Regularization and normalization are two different things!
2. The writing of the paper is extremely poor. The current draft is full of trivial equations that nobody can follow. For signal processing experts, the propositions can be easily derived in one or two lines (I will show you some examples below); and for non-experts, the equations are too tedious to follow. Even if these equations were important, they should go to appendix instead of distracting the readers.

Now let's talk about the technical issues.
1. For Section 3, there is no mathematical difference between convolution and cross-correlation. For convolutional layers, you can simply interpret that the coefficients are stored in a reversed manner, and the layer is indeed a standard convolution. Making the distinction between them only makes the draft unnecessarily hard to understand.
2. For Section 4.1, the analysis is trivial and can be derived using convolution theorem (convolution in time/spatial domain is equivalent to multiplication in frequency domain) and Parseval's theorem (inner product in time/spatial domain is equivalent to inner product in frequency domain). Following standard notation, let's use lower case letters for time/spatial domain and upper case letters for frequency domain (a \circ b is convolution of two vector, a b is the element-wise product of two vectors).
    - 2a. Equation 6 can be proved in one line: <f1, f2> = <F1, F2> = <K1 X, K2 X> = <K1 K2, X X> = <k1 \circ k2, x \circ x>.
    - 2b. Equation 15/16 is more trivial than you thought: ||k1 \circ k2||^2 = ||K1 K2||^2 = 0 --- it requires all-zero coefficients.
    - 2c. Equation 17: I don't know why the loss can generalize from single channel to multiple channels. But anyway, it is also requiring all-zero coefficients.
In summary, the paper is full of tedious equations trying to prove well-known results.
3. Indeed, the problem that the authors are trying to prove is trivial --- I can easily craft a case to do a proof by contradiction.
    - 3a. Suppose we have two input vectors, x_1 = sqrt(2) * [0.7, 0.1], x_2 = sqrt(2) * [0.7, -0.1]. These two unit-length vectors are not identical, so similarity is less than one.
    - 3b. By Fourier transform, we have X_1 = [0.8, 0.6], X_2 = [0.6, 0.8] in frequency domain.
    - 3c. Suppose the kernel is orthogonal, we can craft a orthogonal transformation such that output vectors are Y_1 = [0.8, 0.8], Y_2 = [0.6, 0.6] in frequency domain (you can verify that such transformation can be completed by an orthogonal convolution).
    - 3d. Therefore, y_1 = sqrt(2) * [0.8, 0], y2 = sqrt(2) * [0.6, 0]. Since these two vectors have the same direction, the similarity INCREASE to one.

---

> ### Author Response · Authors · 2025-03-20
> **Authors' Reply to Reviewer**
>
> *Remark*: Parts of the reviewer's remarks were cut out due to character limitation.
>
> *1-Reviewer:* First of all, kernel orthogonality was NOT supposed to reduce feature map redundancy (and I am not aware of such claim before). The purposes of kernel orthogonality are for robustness, invertibility, and regularization. There is another line of works to reduce feature map redundancy, such as network deconvolution or simply batch/layer normalization [. . . ]
>
> *1-Response:* The claim that kernel orthogonality was not supposed to reduce feature map redundancy is incorrect. Please refer to the following:
> - Abstract of [1] and Section 1 paragraph 2 “When filters are learned to be as orthogonal as possible, they become de-correlated. Their filter responses are much less redundant”.
> - [2] Section 3.2 paragraph 3 “Equation 3 constraints orthogonality among filters in one layer, which makes the learned features have minimum correlation with each other, thus implicitly reduce redundancy and enhance the diversity among the filters”.
> - [4] “Imposing orthogonality on the layers of neural networks is known to facilitate the learning by [. . . ]; decorrelate features”.
>
> *2-Reviewer:* The writing of the paper is extremely poor. The current draft is full of trivial equations that nobody can follow. For signal processing experts, the propositions can be easily derived in one or two lines (I will show you some examples below); and for non-experts, the equations are too tedious to follow. Even if these equations were important, they should go to appendix [. . . ]
>
> *2-Response:* We are working on a revised version according to the comments, with most of the equations moved to Appendix, and are happy to present it to the reviewer when required.
>
> *3-Reviewer:* For Section 3, there is no mathematical difference between convolution and cross-correlation. For convolutional layers, you can simply interpret that the coefficients are stored in a reversed manner, and the layer is indeed a standard convolution.
>
> *3-Response:* The convolution and cross-correlation operations, while similar, are distinct. For example, convolution is commutative mathematically, whereas cross-correlation is not. Although the distinction between the two is often overlooked in the context of Machine Learning, it was important to differentiate them in the theoretical analysis without sacrificing generality. However, we have improved the readability in the revised version with reduced focus on the distinction of the two concepts.
>
> *4-Reviewer:* For Section 4.1, the analysis is trivial and can be derived using convolution theorem (convolution in time/spatial domain is equivalent to multiplication in frequency domain) and Parseval’s theorem (inner product in time/spatial domain is equivalent to inner product in frequency domain). Let’s use lower case letters for time/spatial domain and upper case letters for frequency domain. Equation 6 can be proved in one line [...]
>
> *4-Response:* The authors agree with the reviewer’s comment partially. Using the Convolution Theorem and Parseval’s Theorem offers a simpler approach, utilizing both the time and frequency domains to derive the result. The revised version will present both methods. However, this is only valid for a fixed number (i.e., N-1, where N is the size of kernel) of zero-paddings, as implied by the convolution operation in signal processing. We have also provided the proof and discussion of <N-1 zero-padding case in the paper.
>
> *5-Reviewer:* Equation 15/16 is more trivial than you thought: ||k1 ◦ k2||2 = ||K1K2||2 = 0 — it requires all-zero coefficients.
>
> *5-Response:* In general, the equation is valid. However, the interaction between the two filters through convolution may cancel out, leading to the results toward zero.
>
> *6-Reviewer:* Indeed, the problem that the authors are trying to prove is trivial — I can easily craft a case to do a proof by contradiction.
> [...]
>
> *6-Response:* If we understand this comment correctly, our work actually refutes the orthogonal kernel hypothesis, and provides a kernel convolution (referred to as convolutional similarity in the paper) minimization solution. In addition, we assume the kernels operation the same input at a time. However, if you mean that the problem stated in the title can be refuted by providing a counter-example, we apologize for the incomplete information conveyed by the original title. We have updated the title in the revision to ”Feature Map Similarity Reduction in Convolutional Neural Networks.”
>
> *References:*
>
> [1] Orthogonal Convolutional Neural Networks, Wang et al.
>
> [2] All you need is beyond a good init: Exploring Better Solution for Training Extremely Deep Convolutional Neural Networks with Orthonormality and Modulation, Xie et al.
>
> [3] Can we gain more from orthogonality regularizations in Training Deep CNNs, Bansal et al.
>
> [4] Existence, Stability and Scalability of Orthogonal Convolutional Neural Networks, Achour et al.

---

### Review · Reviewer_Le1x · 2025-01-14

**Summary Of Contributions:**

This paper is about feature map redundancy in CNNs. Feature map redundancy is when there's a correlation between feature map values in a convolutional layer. The authors claim that feature map similarity is importantly different from kernel similarity, and that optimizing to reduce feature map redundancy is better in various ways than optimizing to reduce kernel redundancy.

The core difference between feature map similarity and kernel similarity is that the former is dataset dependent: it measures the similarity of conv neuron outputs on a dataset, rather than measuring their similarity in a dataset independent way.

(I'd like to apologize to the authors and editors for the lateness of my review.)

**Audience:**

Yes

**Claims And Evidence:**

No

**Requested Changes:**

I am confused by some aspects of the results, and would like to request comment on them. In particular, I think that equation 23 is equal to zero if different elements of the X vector are uncorrelated. But in section 4.3, the experiments involve X vectors that have elements that I think are sampled iid. Can the authors explain this? More generally, do you agree with me that the core difference between feature map similarity and kernel similarity is the dataset dependence? If so, can you emphasize this more in the paper?

**Strengths And Weaknesses:**

The basic idea of the paper seems reasonable. The results are moderately compelling, though I'm generally skeptical of this kind of small-scale architecture fiddling leading to important insights.

---

> ### Author Response · Authors · 2025-03-20
> **Authors' Reply**
>
> *1-Reviewer:* This paper is about feature map redundancy in CNNs. Feature map redundancy is when there’s a correlation between feature map values in a convolutional layer. The authors claim that feature map similarity is importantly different from kernel similarity, and that optimizing to reduce feature map redundancy is better in various ways than optimizing to reduce kernel redundancy.
>
> *1-Response:* Yes. The existing literature [1, 2] claims that reducing kernel similarity ultimately minimizes feature map redundancy. However, this research empirically (Section 4.1) and theoretically (Section 4.2) demonstrates that this claim is false: reducing kernel similarity is neither a sufficient nor a necessary condition for minimizing feature map redundancy. Based on this finding, we propose Convolutional Similarity to address feature map redundancy. The logic and clarity of expression in the paper will be improved in the next version.
>
> *2-Reviewer:* The basic idea of the paper seems reasonable. The results are moderately compelling, though I’m generally skeptical of this kind of small-scale architecture fiddling leading to important insights.
>
> *2-Response:* This research both empirically and theoretically demonstrates that reducing kernel similarity is neither a sufficient nor a necessary condition for minimizing feature map redundancy. Empirically, the small-scale architectures fit the dataset well. However, the authors agree with the reviewer that more comprehensive testing with different models could be conducted to evaluate the effect of Convolutional Similarity on model performance.
>
> *3-Reviewer:* I think that equation 23 is equal to zero if different elements of the X vector are uncorrelated. But in section 4.3, the experiments involve X vectors that have elements that I think are sampled iid. Can the authors explain this?
>
> *3-Response:* As shown in the paper, Equation 23 is derived from Equation 13, which reformulates the inner product of two feature maps as the inner product of the cropped auto-correlation of X and the cross-correlation of the two kernels. Equation 23 corresponds to the case where the kernels are decorrelated, and thus their inner product is equal to 0. Yes. In Section 4.3, the elements of X vectors are sampled iid. Two iid data sets are uncorrelated but not necessarily orthogonal (i.e., their inner product is not necessarily zero). Additionally, since X, as the input to the convolutional layer, can take any values, iid sampling is a reasonable choice.
>
> *4-Reviewer:* More generally, do you agree with me that the core difference between feature map similarity and kernel similarity is the dataset dependence? If so, can you emphasize this more in the paper?
>
> *4-Response:* Yes. This research proposes a dataset-independent method called Convolutional Similarity to reduce feature map similarity and redundancy, thereby improving model performance. We will highlight the differences in context.
>
> *References*:
>
> [1] Orthogonal Convolutional Neural Networks, Wang et al.
>
> [2] All you need is beyond a good init: Exploring Better Solution for Training Extremely Deep Con-
> volutional Neural Networks with Orthonormality and Modulation, Xie et al.

---

### Review · Reviewer_jcza · 2025-03-06

**Summary Of Contributions:**

This paper investigates the commonly held belief that kernel orthogonality in Convolutional Neural Networks leads to a reduction in feature map redundancy. The authors challenge this assumption by providing theoretical and empirical evidence that kernel orthogonality does not necessarily decrease feature map similarity and may even increase it in some cases​. To address this issue, they propose a novel loss function called Convolutional Similarity, which explicitly minimizes feature map similarity independently of the input data​. Their experiments demonstrate that reducing Convolutional Similarity improves classification accuracy, accelerates convergence, and enables the use of smaller models without performance degradation.

**Audience:**

Yes

**Claims And Evidence:**

Yes

**Requested Changes:**

**1. Expand Experiments:** Include results on more diverse datasets (e.g., ImageNet) to enhance the generalizability of the findings.

**2. Discuss Practical Trade-offs:** Provide a more detailed discussion of the computational cost versus performance gain trade-off for the Convolutional Similarity loss, helping practitioners weigh its benefits against overhead.

**3. Small suggestion:** I strongly suggest move most of the equations to appendix and only keep these important conclusions in the main paper which would be more clear.

**Strengths And Weaknesses:**

**Strengths**

**1. Novelty:** The paper provides a fresh perspective on the relationship between kernel orthogonality and feature map redundancy, and it challenges a widely accepted assumption. Although I am not fully convinced by by this but it indeed provide some inspiration to the community.

**2.Writing and theoretical Part:**  Overall, the paper is well-organized, with a logical progression from problem identification to theoretical analysis, methodology, and experiments. The theoretical analysis in this paper is rigorous and looks good to me.  It is easy to follow and understand.

**3. Practical Implications:** The paper has great implications for improving CNN efficiency – using the new loss leads to more efficient capacity use, allowing smaller models to achieve high performance (comparable to larger models)​. I would love to see how it can take effect in transformer-based model In the era of LLM.

**Weakness:**

**1. Limited Dataset:** Most experiments are conducted on CIFAR10. It is too old. It would strengthen the claims to include results on larger-scale datasets like ImageNet.

**2. Computation Overhead:** This method may introduces additional computational costs.  It may be a limiting factor for large-scale applications.

---

> ### Author Response · Authors · 2025-03-20
> **Authors' Reply to Reviewer**
>
> *1-Reviewer:* Weakness-1. Limited Dataset: Most experiments are conducted on CIFAR10. It is too old. It would strengthen the claims to include results on larger-scale datasets like ImageNet. Weakness-2. Computation Overhead: This method may introduces additional computational costs. It may be a limiting factor for large-scale applications.
>
> *1-Response:* We are happy to carry out further experiments on larger-scale datasets, such as ImageNet, using the same process applied to CIFAR-10 and compare the results. During the experiments, the computational cost will be thoroughly evaluated and compared with the vanilla CNN approach.
>
> *2-Reviewer:* Requested Changes-1. Expand Experiments: Include results on more diverse datasets (e.g., ImageNet) to enhance the generalizability of the findings.
>
> *2-Response:* Answered in Response to Weakness. We will identify representative datasets in the domain for thorough testing.
>
> *3-Reviewer:* Requested Changes-2. Discuss Practical Trade-offs: Provide a more detailed discussion of the computational cost versus performance gain trade-off for the Convolutional Similarity loss, helping practitioners weigh its benefits against overhead.
>
> *3-Response:* The trade-off between computational cost and performance gain will be evaluated during experiments using various datasets and metrics, such as training cost, inference cost, and performance indicators (e.g., accuracy and robustness).
>
> *4-Reviewer:* Requested Changes-3. Small suggestion: I strongly suggest move most of the equations to appendix and only keep these important conclusions in the main paper which would be more clear.
>
> *4-Response:* The authors agree with the reviewer’s comment. We have started to simplify the deduction process, including introducing theorems, moving most of the equations to the appendix and retaining only key deductive procedures and conclusions in the main paper. Thank you!

---

### Decision · Action_Editor_KpZf · 2025-05-15

**Recommendation:** Reject

**Comment:**

The reviewers generally appreciated the attempt to revisit assumptions around kernel orthogonality and feature map decorrelation. Reviewer jcza found the idea well-motivated and the derivations mostly clear. The notion of explicitly optimizing feature similarity is conceptually appealing, and the authors responded to concerns by revising the mathematical exposition and cleaning up the derivations.

However, significant concerns remain. Several reviewers questioned the framing of the problem, pointing out that the connection between kernel orthogonality and feature decorrelation is not as widely accepted as suggested. Moreover, the theoretical claims are largely straightforward derivations from standard properties (e.g., Parseval's theorem), limiting the strength of the contribution. The experimental validation is confined to CIFAR-10 and shallow networks, leaving doubts about practical relevance. Most importantly, although the authors promised to include ImageNet results in the revised draft, those experiments are not present in the final submission, weakening confidence in the paper’s empirical thoroughness. Finally, the writing remains dense and somewhat inaccessible, particularly for readers unfamiliar with signal processing concepts.

Considering these aspects, the AE recommends rejection from TMLR.

**Audience:**

The general topic of feature map redundancy and model efficiency in convolutional networks is of interest to the TMLR audience, particularly those working on neural network architecture, interpretability, and training optimization. However, due to the limited scope of experiments and lack of validation on modern architectures or datasets, the appeal may be narrow and primarily relevant to researchers focused on theoretical aspects of CNN training dynamics rather than broader applied audiences.

**Claims And Evidence:**

The paper proposes a method for reducing feature map redundancy in CNNs by directly minimizing feature similarity rather than relying on kernel orthogonality. The authors argue that kernel orthogonality does not reliably reduce feature redundancy and support their claim with theoretical derivations and empirical tests. Two strategies—regularization during training and iterative initialization—are proposed and evaluated using small CNNs and ResNet-18 on CIFAR-10.

The paper makes two major claims: 1) kernel orthogonality does not reliably reduce feature map similarity in CNNs, and 2) explicitly minimizing feature map similarity using the proposed method leads to improved performance and more efficient model capacity use.

The first claim is supported by theoretical and empirical analysis. The theoretical derivations are mathematically correct but mostly straightforward applications of known signal processing results (e.g., Parseval’s theorem). The empirical evidence provided (e.g., cases where orthogonal kernels yield similar feature maps) supports the basic insight.

The second claim is only partially supported. While the authors demonstrate performance gains on CIFAR-10 using small CNNs and ResNet-18, they do not test on larger-scale datasets or modern architectures. Despite promising to include ImageNet results in the revision, these were not added. This limits the strength of the evidence regarding the method’s generalizability. Additionally, the paper lacks a detailed analysis of trade-offs such as computational overhead.